# Royal Jelly and Its Components Promote Healthy Aging and Longevity: From Animal Models to Humans

**DOI:** 10.3390/ijms20194662

**Published:** 2019-09-20

**Authors:** Hiroshi Kunugi, Amira Mohammed Ali

**Affiliations:** 1Department of Mental Disorder Research, National Institute of Neuroscience, National Center of Neurology and Psychiatry, Tokyo 187-8551, Japan; hkunugi@ncnp.go.jp; 2Department of Psychiatric Nursing and Mental Health, Faculty of Nursing, Alexandria University, Alexandria 21527, Egypt

**Keywords:** aging, alternative therapy, composition of royal jelly, dietary interventions, healthspan, lifespan, longevity, royal jelly, *IGF-1*, oxidative stress

## Abstract

Aging is a natural phenomenon that occurs in all living organisms. In humans, aging is associated with lowered overall functioning and increased mortality out of the risk for various age-related diseases. Hence, researchers are pushed to find effective natural interventions that can promote healthy aging and extend lifespan. Royal jelly (RJ) is a natural product that is fed to bee queens throughout their entire life. Thanks to RJ, bee queens enjoy an excellent reproductive function and lengthened lifespan compared with bee workers, despite the fact that they have the same genome. This review aimed to investigate the effect of RJ and/or its components on lifespan/healthspan in various species by evaluating the most relevant studies. Moreover, we briefly discussed the positive effects of RJ on health maintenance and age-related disorders in humans. Whenever possible, we explored the metabolic, molecular, and cellular mechanisms through which RJ can modulate age-related mechanisms to extend lifespan. RJ and its ingredients—proteins and their derivatives e.g., royalactin; lipids e.g., 10-hydroxydecenoic acid; and vitamins e.g., pantothenic acid—improved healthspan and extended lifespan in worker honeybees *Apis mellifera*, *Drosophila Melanogaster* flies, *Gryllus bimaculatus* crickets, silkworms, *Caenorhabditis elegans* nematodes, and mice. The longevity effect was attained via various mechanisms: downregulation of insulin-like growth factors and targeting of rapamycin, upregulation of the epidermal growth factor signaling, dietary restriction, and enhancement of antioxidative capacity. RJ and its protein and lipid ingredients have the potential to extend lifespan in various creatures and prevent senescence of human tissues in cell cultures. These findings pave the way to inventing specific RJ anti-aging drugs. However, much work is needed to understand the effect of RJ interactions with microbiome, diet, activity level, gender, and other genetic variation factors that affect healthspan and longevity.

## 1. Introduction

Reduced mortality from both communicable and noncommunicable diseases accelerates the growth of aging populations worldwide. However, long living entails reduced healthspan out of increased burden of noncommunicable diseases [1]. Aging, healthspan, and lifespan are tightly related concepts. Though aging is a natural phenomenon, numerous factors such as stress, poor nutrition, and pollution are associated with increased internal production of free radicals, which enhance chronic subclinical inflammation and lead to faster aging [2,3]. Aging-related oxidative damage and inflammation contribute to a trail of cellular and molecular alterations: telomere attrition, epigenetic alterations, genome instability, reduced proteostasis, disturbed nutrient sensing, mitochondrial dysfunction, cellular senescence, stem cell exhaustion, and altered intercellular communication. In this respect, aging represents a process of random cellular degradation [4,5,6], which is associated with increased likelihood of progressive loss of function, decreased fertility, and increased risk of various diseases (both physical and cognitive)—contributing to premature death and increased morbidity in aging populations (Figure 1) [2,7].

A recent review indicates that family profiles of centenarians emphasize the contribution of “protective or longevity genes” to extreme longevity as well as to low incidence of age-related diseases in this group [5]. Apart from genetic factors, favorable environmental conditions and healthy lifestyles contribute to low morbidity in people who survive to extreme ages [8,9]. Therefore, the worldwide substantial increase of aging populations, who encounter a trail of debilitating illnesses, has driven researchers to find strategies to slow aging and support a healthy lifespan by delaying the onset of age-related diseases [5]. Research denotes that the natural decline of function that occurs with aging, as well as age-related disorders, can be prevented and/or reversed through environmental modifications such as dietary interventions [10,11]. Accumulating evidence indicates that functional foods can promote healthy aging, reduce morbidity, and lengthen healthy lifespans [12]. Royal jelly (RJ) is considered a functional food (with a documented safety profile) because it has a range of pharmacological activities: antioxidant, anti-inflammatory, antitumor, antimicrobial, anti-hypercholesterolemic, vasodilative, and hypotensive. Indeed, RJ has been widely used to treat several health conditions such as diabetes mellitus, cardiovascular diseases, and cancer, to name a few [13,14].

RJ is a creamy substance secreted from the mandibular and hypopharyngeal glands of the worker honeybee *Apis mellifera*. It is the food of all bee larvae during their first 3 days of life; workers then shift to worker jelly (WJ), composed mainly of honey and pollen, while bee queens continue to consume RJ [15,16,17]. It is suggested that RJ is a potent promoter of healthy aging and longevity because it enhances overall health and fertility of queen bees, who may lay up to 3000 eggs a day and survive as long as five years compared with infertile workers that live up to 45 days only [18,19,20]. However, studies of the effect RJ on longevity in humans are scarce, presumably due to technical and ethical reasons. In this respect, this paper reviews, synthesizes, and discusses the most relevant studies that used RJ to enhance healthspan and extend lifespan in different species. We also examined the metabolic, molecular, and cellular mechanisms associated with the longevity-promoting properties of RJ.

## 2. Chemical Composition of RJ

Water constitutes 60–70% (*w*/*w*) of fresh RJ; pH of fresh RJ usually ranges between 3.6 and 4.2. The various pharmacological properties of RJ are attributed to its unique and rich composition of proteins, carbohydrates, vitamins, lipids, minerals, flavonoids, polyphenols, as well as several biologically active substances [21]. Herein, we provide a detailed description of most ingredients of RJ.

### 2.1. Sugars

Sugars comprise 7.5–15% of RJ content. Fructose and glucose constitute 90% of the total sugar fraction of RJ, whereas sucrose accounts for 0.8–3.6%. RJ contains very small amounts of other sugars such as maltose, trehalose, melibiose, ribose, and erlose [22]. RJ sugar content varies remarkably from one sample to another based on season, geographical location, botanical origin, bee species, and method of production. For example, French RJ contents of sucrose and erlose are less than 1.8% and 0.4%, respectively, whereas their concentrations in RJ produced by sugarcane feeding are comparatively higher (7.7% and 1.7%) [23]. Sugars of RJ are thought to contribute to its epigenetic effects, given that RJ sugar content is extremely high compared with WJ (the main food of bee workers). Meanwhile, supplementing WJ with fructose and glucose (4%) fosters the development of larvae to adult workers. In the same way, a gradual increase of WJ sugar concentrations (up to 20%) increases larval consumption of WJ and eventually results in the development of intercastes (midway between queens and workers) and queens at a rate similar to that obtained by pure RJ for in vitro rearing [24,25]. Thus, sugars of RJ represent a phagostimulant that functions through the *insulin/insulin-like* signaling cascades and the nutrient sensing *mTOR* pathway to derive larval development by increasing quantities of ingested food and increasing intake of nutrients necessary for queen development [25].

### 2.2. Lipids

Among the principal bioactive constituents of RJ, lipids constitute 7–18% of RJ content; 90% of these lipids are rare and unique short hydroxy fatty acids with 8–12 carbon atoms in the chain and dicarboxylic acids. The most prominent RJ fatty acids in order are 10-hydroxydecanoic acid (10-HDA), 10-hydroxy-2-decenoic acid (10H2DA), and sebacic acid (SA) [18]. The 10-HDA exerts epigenetic control over caste differentiation of *Apis mellifera* by inhibiting histone deacetylases, which catalyze the hydrolysis of ε-acetyl-lysine residues of histones [25,26]. Due to the low pH of RJ, 10-HDA acts as a strong bactericidal. It therefore, protects bee larvae against virulent bacterial infections that affect bee hives such as those caused by some strains of *Paenibacillus* larvae [27]. In mammals, it protects mice against pulmonary damage induced by lipoteichoic acid, a toxin from *Staphylococcus aureus* [28]. It also exerts an antibacterial effect against various pathogenic bacterial species in human cancer colon cells [29]. The 10-HDA may be used to treat age-related neurodegenerative disorders given its documented neurogenic activity—it stimulates neuronal differentiation from progenitor cells (PC12) cells through mimicking the effect of brain-derived neurotrophic factor [30]. In addition, it possesses neuroprotective effects against glutamate- and hypoxia-induced neurotoxicity [31]. The 10-HDA may also be used for manufacturing cosmetics and anticancer drugs, given that it increases skin-whitening and exerts antiproliferative effects on B16F10 melanoma cells by inhibiting the expression of microphthalmia-associated transcription factor and tyrosinase-related protein 1 (*TRP-1*) and *TRP-2* [32]. Moreover, 10-HDA has been identified as an inhibiting factor of matrix metalloproteinases (*MMPs*)—proenzymes activated by proteolytic cleavage under inflammatory conditions, which degrade matrix and non-matrix proteins and contribute to tissue aging (e.g., skin) and cause serious disabling diseases such as rheumatoid arthritis [33,34].

SA, 10-HDA, and 10H2DA, demonstrate anti-inflammatory effects through regulation of several proteins involved in the mitogen-activated protein kinase (*MAPK*) and nuclear factor kappa-B signaling [29,35]. Moreover, these acids mediate estrogen signaling by enhancing the activity of estrogen receptors (*ERs*) *ER*α, *ERβ* [36], which can benefit bone, muscle, and adipose tissue in a sex-dependent manner [31]. A derivative of 4-hydroperoxy-2-decenoic acid known as 4-hydroperoxy-2-decenoic acid ethyl ester (HPO-DAEE) prevents 6-hydroxydopamine-induced cell death in human neuroblastoma SH-SY5Y cells through triggering slight emission of reactive oxygen species (ROS), which stimulates the production of antioxidants via activation of antioxidant pathways: nuclear factor erythroid 2 (*NRF2*)-antioxidant response element (*ARE*) and eukaryotic initiation factor 2 (*eIF2α*), an upstream effector of the activating transcription factor-4 (*ATF4*) [37]. HPO-DAEE also demonstrates anticancer effects through accumulation of intracellular ROS and activation of proapoptotic CCAAT-enhancer-binding protein homologous protein expression [38]

### 2.3. Proteins

Proteins are the dominant ingredient of RJ (50% of its dry matter) and more than 80% of total RJ proteins are composed of nine major RJ proteins (MRJPs, 49–87 kDa)—the first five MRJPs constitute up to 82–90% of MRJPs. Glycosylation and phosphorylation of MRJPs is essential for biological processes that involve glycoproteins, such as cell adhesion, cell differentiation, cell growth, and immunity. MRJPs modulate the development of female larvae, not only through their high nutritional value but mainly through physiological activity of their highly homologous 400–578 amino acids that contribute to RJ’s role in cell proliferation, cytokine suppression, and antimicrobial activity [12].

Research documents anti-senescence activity of MRJPs for human cells in vitro [39]. MRJP1 is the most dominant among all MRJPs; essential amino acids constitute 48% of its content. Circular dichroism measurements indicate that the secondary structure of MRJP1 consists of 9.6% α-helices, 38.3% β-sheets, and 20% β-turns [22]. MRJP1 exists in two distinct forms: oligomer and monomer. Oligomer MRJP1 is highly heat-resistant and it is considered a predominant proliferation factor compared with MRJP2 and MRJP3 [40]. High-performance liquid chromatography and SDS-PAGE analyses of MRJP1 revealed the presence of a 57 kDa monomeric glycoprotein, which can be degraded at 40 °C, known as royalactin. Royalactin mimics the effect of epidermal growth factor (*EGF*) in rat hepatocytes and modulates the development of bee larvae [41]. Royalactin is reported to bind with the most sensitive regions in mouse embryonic stem cell culture, resulting in activation of a pluripotency gene network that enables self-renewal of stem cells [42]. MRJP1 exerts nematicidal activity against *C. elegans* via constant downregulation of a rate-controlling enzyme of the citric acid cycle known as isocitrate dehydrogenase encoding the idhg-1 gene [43]. On the other side, MRJP2 and its isoform X1 exhibit potent anticancer effects and protect hepatocytes against CCl4 toxicity by inducing caspase-dependent apoptosis, scavenging intracellular free radicals, inhibiting tumor necrosis factor (*TNF*)-α, and mixed lineage kinase domain-like protein [44].

RJ contains proteins other than MRJPs, albeit in small amounts, such as royalisin, jelleines, and aspimin. Royalisin and jelleines are common RJ antimicrobial peptides that enhance efficiency of the immune response of bee larvae to various infections. The structure of royalisin is highly compact due to its high cysteine content, which boosts its stability at low pH and high temperature. On the other hand jelleines are thought to stem from trypsin digestion of MRJP1 by the action of exo-proteinase of the hypopharyngeal glands on C-terminal to N-terminal tryptic fragment. Peptides of royalisin and jelleines possess hydrophobic residues, which contribute to their antimicrobial properties by affecting functions of bacterial membranes. RJ also contains apolipophorin III-like protein, a lipid binding protein that exerts antimicrobial effect by carrying lipids into aqueous environments through the formation of protein–lipid complexes. In addition, the antibacterial effect of RJ is partially attributed to its glucose oxidase enzyme, which catalyzes the oxidation of glucose to hydrogen peroxide [45]. Recently, examination of whole RJ and single protein bands by off-line LC-MALDI-TOF MS glycomic analyses, complemented by permethylation, Western blotting, and arraying data, revealed the presence of glucuronic acid termini, sulfation of mannose residues, core β-mannosylation of the N-glycans, and a fairly scarce zwitterionic modification with phosphoethanolamine, which may contribute to the development of honey bees and their innate immunity [46].

### 2.4. Phenols, Flavonoids, and Free Amino Acids

The antioxidant potency of RJ, at least in part, is attributed to its polyphenolic compounds and flavonoids, which are measured based on gallic acid and rutin equivalent (GAE and RE), respectively [47]. RJ contains 23.3  ±  0.92 GAE µg/mg and 1.28  ±  0.09 RE µg/mg of total phenolics and flavonoids, respectively. The vast phenolic content of RJ consists of pinobanksin, organic acids (e.g., octanoic acids, dodecanoic acid, 1,2-benzenedicarboxylic acid), and their esters. Meanwhile, flavonoids of RJ can be differentiated into four groups: (1) flavanones e.g., hesperetin, isosakuranetin, and naringenin; (2) flavones e.g., acacetin, apigenin and its glucoside, chrysin, and luteolin glucoside; (3) flavonols e.g., isorhamnetin and kaempferol glucosides; and (4) isoflavonoids e.g., coumestrol, formononetin, and genistein. The antioxidant activity of these components contributes to the antiapoptotic and anti-inflammatory properties of RJ [18]. Age of larvae that produce RJ (1–14 days) and harvesting time affect RJ content of phenols and amino acids; RJ harvested from the youngest larvae (one-day-old) within 24 h contains higher proteins and polyphenolic compounds and exhibits stronger free radical scavenging effect compared with RJ harvested from older larvae or later than 24 h [47]. Small peptides such as di-peptides (Lys-Tyr, Arg-Tyr, and Tyr-Tyr) obtained from RJ proteins hydrolyzed by protease N possess high antioxidative activity owing to their phenolic hydroxyl groups, which scavenge free radicals by releasing a hydrogen atom [48].

RJ is rich in amino acids, including essential ones [22]. Concentrations of free amino acids in RJ increase with harvesting time from 4.30 mg/g at 24 h to 9.48 mg/g at 72 h, whereas levels of total amino acids decrease by time from 197.96 mg/g at 24 h to 121.32 mg/g at 72 h [49]. LC/MS method analysis and hydrophilic interaction liquid chromatography-tandem mass spectrometry indicate that lysine is the most prominent free amino acid in RJ (62.43 mg/100 g), followed by proline (58.76 mg/100 g), cystine (21.76 mg/100 g), and aspartic acid (17.33 mg/100 g). RJ contains less than 5 mg/100 g of other amino acids such as valine, glutamic acid, serine, glycine, cysteine, threonine, alanine, tyrosine, phenylalanine, hydroxyproline, leucine-isoleucine, and glutamine [18,50].

The amino acid content in protease-treated RJ (pRJ)—obtained by removal of two allergen proteins to convert RJ hydrolysates into shorter chain monomers that are easy to absorb—is greater than in crude RJ [51]. Moreover, eluted water pRJ with 30% MeOH contains higher levels of dipeptides and tripeptides than pRJ. These components are likely to possess a lifespan-prolonging activity since pRJ increases lifespan of *C. elegans* compared with RJ [52]. Evidence signifies that amino acids of RJ can prolong lifespan in mammals. Long-term dietary supplementation of branched-chain amino acid-enriched mixture (BCAAem), which contains 3 branched-chain amino acids that can be found in RJ (leucine, isoleucine, and valine), enhances mitochondrial biogenesis and sirtuin 1 expression, and reduces ROS production in cardiac and skeletal muscle, which is associated with alleviation of age-related muscle dysfunction, resulting in an increase of the average lifespan of male mice. The effect of these amino acids on mitochondrial biogenesis involved activation of signaling pathways of endothelial nitric oxide synthase (*eNOs*) and *mTOR* and its substrates: *S6K* and eukaryotic translation initiation factor (*eIF4E*)-binding protein (*4E-BP1*) [53].

### 2.5. Vitamins, Minerals, and Bioactive Substances

Pantothenic acid (vitamin B5) is the most abundant vitamin in RJ (52.8 mg/100 g), followed by niacin (42.42 mg/100 g). RJ contains small amounts of various B group vitamins (B1, B2, B6, B8, B9, and B12), ascorbic acid (vitamin C), vitamin E, and vitamin A [18,22]. Gardner (1948) noted that pantothenic acid of RJ is a lifespan-extending agent [54].

Mineral salts constitute 1.5% of RJ content [18]. Inductively coupled plasma optical emission spectroscopy and double focusing magnetic sector field inductively coupled plasma mass spectrometry indicate that RJ contains small amounts of various minerals and trace elements such as K, Na, Mg, Ca, P, S, Cu, Fe, Zn, Al, Ba, Sr, Bi, Cd, Hg, Pb, Sn, Te, Tl, W, Sb, Cr, Mn, Ni, Ti, V, Co, and Mo. Whereas concentrations of trace and mineral elements in honey vary according to botanical origin, RJ content of trace elements and minerals is highly constant. In this respect, RJ can be considered a form of larval lactation that possesses homeostatic adjustment, same as mammalian and human breast milk [55].

RJ contains high amounts of acetylcholine (Ach, 4–8 mM), and RJ concentration of Ach is highly conserved because of its acidic pH [56]. It is well-known that Ach acts as a neurotransmitter that plays a major role in memory formation and cognitive functioning. Glucose metabolism and insulin contribute to Ach synthesis by controlling the activity of choline acetyltransferase [57]. In this respect, consumption of RJ may prevent the development of cognitive dysfunction thanks to its Ach content. In addition, Ach content of RJ has a survival promoting effect [56].

RJ is rich in nucleotides such as free bases (e.g., adenosine, uridine, guanosine, iridin, and cytidine) and phosphates (e.g., adenosine diphosphate (ADP), adenosine triphosphate (ATP), and adenosine monophosphate (AMP)). Nucleotides constitute 2682.9 mg/kg and 3152.8 mg/kg in fresh and commercial RJ, respectively. Levels of ADP, ATP, and AMP are higher in fresh RJ, and therefore, they can signify RJ freshness. These compounds are necessary for organisms’ physiological activities e.g., metabolic degradation of intracellular ATP is essential to provide cells with energy necessary for transport systems and enzymatic activities of proteins [22,58]. Among all nucleotides, AMP N1-oxide is considered a unique active component that exists nowhere in nature except in RJ. AMP N1-oxide demonstrates neurogenic and neurotrophic activities: it stimulates neurite outgrowth and induces differentiation of PC12 cells into neurons similar to sympathetic neurons. This action is similar to that of nerve growth factor, which functions through activation of two cascades of cellular signaling *MAPK*/extracellular signal-regulated kinase 1 or 2 (*ERK1/2*) and phosphatidylinositol 3-kinase/*Akt* pathways. The neurite outgrowth-promoting activity of AMP N1-oxide is mediated by adenylate cyclase-coupled adenosine A2A receptors, which are highly expressed in the brain (striatum in particular)—adenosine A2A receptors prevent radical formation and apoptosis, and they contribute to early neuronal development and regulation of synaptic plasticity [18,59].

## 3. Healthspan and Longevity Effects of RJ in Various Species

### 3.1. RJ Enhances Fertility and Longevity in Population of the Beehive

Numerous bioactive elements are abundant in RJ, making it an optimal food and a well-balanced nutrient-rich diet [25] that, when consumed by bee larvae, induces their development into queens. Meanwhile, larvae that consume honey or pollen grow into workers, which live shorter and are unable to reproduce. On the other hand, emerging bee workers fed RJ-rich diet acquire queen-like morphogenic characteristics such as increased body size and ovary development [41,60,61]. Real-time RT-PCR and HPLC-ECD analysis denoted that augmentation of fertility (ovarian development) in bee workers by RJ is the result of enhancement of brain levels of tyrosine, dopamine, and tyramine by RJ [61], as well as activation of epidermal growth factor receptor (*EGFR*), which increases the production of the juvenile hormone, titre, known to regulate growth [41]. RJ also improves the memory of bee workers [62] and increases their survival [15,41,56,62,63], owing to its high Ach concentration [56], MRJP1 content of royalactin [41], MRJPs 2, 3, and 5 [25], and a water soluble RJ protein extracted by precipitation with 60% ammonium sulfate (RJP_60_) [15].

### 3.2. RJ Enhances Healthspan and Longevity in Other Species

Based on the assumption that RJ is the main factor contributing to long survival of bee queens, several studies examined the lifespan-expanding effects of RJ in diverse species (Figure 2), and results seem to be consistent with the naturally occurring model of bee queens.

#### 3.2.1. Drosophila Melanogaster (Fruit Fly)

The fruit fly *Drosophila melanogaster* (*M.*) has been widely used in the literature as an invertebrate model to understand the pathology of numerous diseases and to examine the effect of various agents, including RJ. The earliest study of the longevity effect of RJ dates back to 1948. That study reported that large doses of dehydrated RJ, its pantothenic acid, as well as its water soluble and insoluble organic acids significantly extended lifespan of *Drosophila M.* compared with controls. The author suggested that pantothenic acid of RJ may have an anti-aging effect by itself or by synergizing the action of other vitamins [54]. In a relatively recent study, RJ (0.1, 0.3, and 0.5 g) significantly increased the average lifespan of both male and female flies—the effect was attributable to enhancement of the antioxidation capacity noticed in flies treated by high doses of RJ: increased superoxide dismutase (SOD) and catalase levels [64].

As a replication of his experiment of RJ and royalactin in bees, Kamakura (2011) treated *Drosophila Canton-S.* larvae with royalactin and 20% fresh RJ. RJ treatment increased body size, cell size, and fertility; prolonged lifespan; and shortened developmental time from larva to adult compared with control. Same as in bees, RT-PCR and enzyme immunoassay showed that both RJ and royalactin induced *EGFR* signaling (not insulin signaling), which activated *S6K* in the fat body, which further increased the body size. *EGFR* also activated the *MAPK* pathway, which increased the synthesis of a biologically active ecdysteroid known as 20-hydroxyecdysone (*20E*) and juvenile hormone—an effect that was demonstrated by growth regulation—reducing the developmental time [41]. A subsequent study stated that supplementing *Drosophila M.* with 1% freeze-dried RJ (FDRJ) powder shortened developmental time, prolonged the lifespan of adult males, and increased females’ egg production without any morphological changes. In female flies, RT-PCR indicated that FDRJ was significantly associated with heightened gene expression of an insulin-like peptide known as dilp5, its insulin receptor (*InR*), and the nutrient sensing molecule *mTOR*, the mechanistic target of rapamycin—all these molecules are known to affect growth and reproduction. However, stimulation of the *insulin/TOR* signaling pathways was not associated with extension of lifespan of FDRJ-fed female flies [17]. Another study noted that supplementation of MRJPs, especially MRJP 1 and MRJP3, at an optimal dose of 2.5% (*w*/*w*) of diet significantly lengthened the mean lifespan of both male and female *Drosophila*. The longevity effect of MRJPs was positively associated with increased feeding and fertility. Microarray data and gene ontology enrichment analyses revealed that the molecular mechanism underlying increased lifespan and fertility was similar to that discovered by Kamakura (2011): MRJP supplementation upregulated the gene expression of *S6K*, *MAPK*, and *EGFR* in *EGFR*-mediated signaling. In addition, MRJPs improved the anti-oxidation capacity of flies by increasing the expression of *CuZn-SOD* gene i.e., SOD levels were higher, while malonaldehyde (MDA) levels were lower than control flies [12]. Another study replicated the size/growth enhancing effect of RJ at low concentrations (10–30%) in *Drosophila M.* [16]. However, males’ lifespan decreased by 20% RJ treatment whereas none of these concentrations affected the lifespan of females. Findings provided no evidence of RJ activation of insulin and *EGFR* signaling pathways, yet RJ regulated the gene expression related to oxidative stress and catabolism. The authors attributed the discrepancy noted between their results and findings of other studies to employing a slightly different strain of *Drosophila* (*Canton-S*), using a commercially available source of RJ, and difference in the nutrient contents of the control culture medium. On the other hand, higher concentrations (40–70%) of RJ had adverse effects: prolonged development time, shortened lifespan, increased mortality, and reduced productivity in both sexes. Data on global gene expression indicated that excess nutrients in high doses of RJ altered cellular processes as a result of altering genes involved in amino acid metabolism and encoding glutathione S transferases, which detoxify xenobiotic compounds [16].

#### 3.2.2. Gryllus Bimaculatus Cricket and Silkworms

The two-spotted cricket *Gryllus (G.) bimaculatus*, a member of primitive group *Polyneoptera*, is another kind of species that were used to examine the longevity effect of RJ. RJ dietary supplementation to *G. bimaculatus* during early nymph stage significantly decreased developmental time, extended lifespan, and increased the body size of both males and females in a dose-dependent manner (8−15% *w*/*w*) compared with the control high protein, sugar, and lipid diet. The prolonged lifespan of RJ was not due to an extended nymph stage since the adult stage emerged earlier in crickets fed RJ than those fed the control diet. Similarly, RJ administration increased body size and egg size in female silkworms. The authors concluded that the effects of RJ were not attributed to the nutritional supplement itself; however, they did not examine the molecular mechanism behind them [65].

#### 3.2.3. Caenorhabditis (C.) Elegans Nematodes

*Caenorhabditis (C.) elegans* nematodes have been used in several instances to test the longevity-promoting activity of RJ. Japanese researchers found that RJ, protease-treated RJ (pRJ), pRJ-Fraction 5 (pRJ-Fr.5), and a derivative of pRJ-Fr.5—10-HDA (the main lipid of RJ)—extended the lifespan of *C. elegans.* RJ 10 µg/mL was an optimal dose for enhancing longevity by 7−9%, while RJ 1 or 100 µg/mL had no effect. Meanwhile, all pRJ concentrations significantly prolonged the mean lifespan—though the greatest effect was noticed at 10 µg/mL, which increased mean lifespan by 7–18%. The longevity effect of combined pRJ-Fr.5 and 10-HDA was greater than that induced by each treatment on its own. DNA microarray and RT-PCR showed that the longevity-promoting effect was attributed to reduction of the *insulin/IGF-1* signaling—pRJ-Fr.5 upregulated the expression of dod-3 gene and downregulated the expression of *ins-9*, an insulin-like peptide gene, along with *dod-19*, *dao-4*, and *fkb-4* genes (further details are shown below in the mechanism section) [52]. In two subsequent studies RJ, pRJ, and 10-HDA enhanced longevity and increased stress resistance of *C. elegans* against thermal, irradiation, and oxidative stress [66,67]. Intact, deglycosylated, and mildly heat-treated royalactin extended mean lifespan of *C. elegans* by 18–34%—higher concentrations produced the vastest lifespan-extending effect. Royalactin also enhanced locomotion, which indicates promotion of healthy aging [68].

#### 3.2.4. Mice

Few studies investigated the lifespan extending activity of RJ in mice. In an early study, intermediate and high doses (50 and 500 ppm) of RJ significantly prolonged the mean lifespan of C3H/HeJ mice by 25%, whereas RJ at a low dose (5 ppm) yielded no significant effect. Meanwhile, all doses of powdered RJ, contrary to bee and *Drosophila* studies, had no effect on mice growth, food intake, or appearance compared with control mice. RJ treatment significantly lowered kidney DNA and serum levels of 8-hydroxy-2-deoxyguanosine, a marker of oxidative stress that increases with aging [2]. Similarly, long-term intragastric administration of RJ and pRJ to a d-galactose-induced aging mice model resulted in numerous anti-aging and healthspan effects: preventing aging-related weight loss, improving memory and motor performance, and delaying aging-related atrophy of thymus, thus preventing diminution of the immune function compared with control animals. The effects were attributed to inhibition of lipid peroxidation and improvement of levels of antioxidant enzymes [69]. Likewise, dietary supplementation of RJ and pRJ (0.05% or 0.5%, *v*/*v*) to genetically heterogeneous head tilt mice—which exhibit vestibular dysfunction, imbalanced position, and inability to swim—could not prolong lifespan but significantly delayed age-related impairment of motor functions, positively improved physical performance of treated mice on four tests (grip strength, wire hang, horizontal bar, and rotarod), lowered age-related muscular atrophy, increased markers of satellite cells (muscle stem cells), and suppressed catabolic genes [70]. Another study examined the survival-expanding time (not lifespan extension) of oral RJ treatment (75, 150, and 300 mg/kg body wt/day for 13 consecutive days) following NaNO2 intraperitoneal injection or decapitation as models of brain hypoxia and complete brain ischemia [71]. Findings indicated that the intermediate dose of RJ (150 mg) significantly expanded survival time, whereas RJ 75 mg had no significant effect, meanwhile RJ 300 mg significantly decreased survival time. The author suggested that low pH of RJ (3.6 to 4.2) in mice treated with high doses of RJ induced activation of acid-sensing channels, increased acidosis of extracellular fluid, and aggravated brain ischemia [71].

## 4. RJ Might Enhance Longevity in Humans by Promoting General Health

It is an important question whether observed effects of RJ on healthspan and longevity in model organisms (e.g., bees, fruit flies, *C. elegans*, silkworms, crickets, and mice) can be generalized to humans. A small number of in vitro studies examined the antiaging activity of RJ, 10-HDA, and MRJPs on human cell lines, and results support findings reported from studies of model organisms. In two experiments, normal human skin fibroblasts were ultraviolet-irradiated (as a model of skin photoaging) and treated with RJ and 10-HDA. Results revealed that RJ and 10-HDA protected cells against ultraviolet A- and B-induced ROS-related oxidative damage, decreased cellular senescence, stimulated the production of procollagen type I and transforming growth factor-β1 [33,72], and suppressed the expression of *MMP-1* and *MMP-3* at transcriptional and protein levels [33]. In another study, human embryonic lung fibroblast cells were treated with different concentrations of MRJPs (0.1–0.3 mg/mL) versus bovine serum albumin. MRJP-treated cells showed the highest proliferation activity, the lowest senescence, and the longest telomeres. Such effects were associated with upregulation of SOD1 and downregulation of *mTOR*, catenin beta like-1, and tumor protein p53 [39]. Apart from these limited in vitro studies, we could not locate any study that investigated the effect of RJ on lifespan in humans in vivo. However, several studies assessed the effect of RJ on promotion of wellbeing and prevention of severe diseases associated with increased early death—these studies may mirror the healthspan effects of RJ. Herein, we explore how RJ may support healthy aging in the general population.

It is becoming clear that certain metabolic pathways reduce longevity in humans by increasing the risk of serious illnesses that contribute to mortality e.g., diabetes mellitus, metabolic syndrome, cardiovascular diseases, and cancer [73]. The life-expanding effect of RJ possibly originates from its antioxidant and anti-inflammatory properties, which can promote healthy aging by improving glycemic status, lipid profiles, and oxidative stress—and hence can prevent the occurrence of various debilitating metabolic diseases [13,14]. In accordance, administration of RJ in healthy volunteers was associated with improved indicators of physical wellbeing (erythropoiesis and glucose tolerance) [74].

Rheumatoid arthritis (RA) is one of the most common disabling disorders that seriously endanger healthspan. It is a chronic systemic inflammatory arthritis that occurs at an age of onset of 55 years and increases with age i.e., it predominantly affects the older population. Meanwhile, treatment is complicated by age-related decline in organ function (e.g., renal and metabolic), comorbidities, and changes in body composition (e.g., decreased lean mass), which make outcomes of RA treatment (steroids and anti-*TNF* agents) in the elderly highly disappointing [75]. On the other hand, a relatively large number of in vitro studies indicate that 10-HDA can be a safe treatment of RA. The 10-HDA is likely to prevent joint destruction by inhibiting *MMP* production from rheumatoid arthritis synovial fibroblasts through blockage of p38 kinase and c-Jun N-terminal kinase-AP-1 signaling pathways [34,76]. The 10-HDA also prevents cell proliferation of fibroblast-like synoviocytes by inhibiting target genes of *PI3K–AKT* pathway and genes of cytokine-cytokine receptor interaction [77].

It is believed that hormones play a major role in healthspan and longevity by regulating cellular responses, metabolism, and growth [78]. Insulin is one of such hormones that affect every single cell in the body; its signaling pathway interacts with a variety of other pathways and affects their functioning [79]. For instance, overexpression of insulin-like growth factors (*IGFs*) and receptors for insulin is associated with increased cell proliferation and risk of cancer [80]. Meanwhile, downregulation of insulin signaling contributes to proper mitochondrial function, suppression of inflammatory mediators, regulation of cellular metabolism, cellular resistance to stress, activation of DNA repair genes, and reduction of oxidative damage of macromolecules and cellular senescence, which all further enhance health and longevity, both in humans and other species [79,81,82]. In this respect, most anti-aging dietary interventions target growth-promoting pathways i.e., they function by downregulating *IGF-1* and *mTOR-S6K* pathway and activating nutrient sensors (*MAPK* and sirtuins), which signal nutrient scarcity and stimulate catabolism [83]. A majority of the above-reviewed studies indicate that RJ enhanced longevity by targeting insulin signaling; this mechanism is likely to work in humans. In fact, RJ has an insulin-like activity—an insulin-like peptide had been purified from RJ and it is similar to insulin of vertebrates in solubility, chromatographic, immunological, and biological characteristics [84]. Administration of RJ to healthy athletes significantly decreased their insulin and increased thyroxine (T4) hormone levels in plasma [85]. In line with this, healthy adults who consumed a single oral dose of RJ (20 g) immediately before oral glucose tolerance test showed significantly reduced glucose level [86].

Aging is usually associated with reduction of sex hormones [87]. The genetic switch model of aging postulates that end of reproduction entails a genetically programmed inactivation of survival and maintenance pathways, which causes a progressive age-related decline of function [88]. In fact, sex hormones are considered markers of longevity since they have a neuroprotective effect, which originates from their ability to improve insulin resistance and enhance the DNA repair capacity of neurons [74,89]. It is becoming clear that RJ modulates sex hormones. The endocrine stimulation exerted by RJ is necessary for ovary development in bee queens and it increased egg-laying in *Drosophila M.* and silkworms [41,63,65]. RJ and royalactin increased juvenile hormone titre (a fertility hormone) downstream of *EGFR* signaling in *Drosophila M.* [12,17,41]. Several studies indicated that RJ and its lipids exert estrogenic activity both by binding with estrogen receptors and activating the expression of endogenous genes [87,90]. Therefore, sex hormones represent another facet through which RJ may enhance longevity. In humans, RJ supplementation to healthy volunteers increased serum testosterone and the ratio of testosterone/dehydroepiandrosterone sulfate, indicating that RJ accelerates conversion of dehydroepiandrosterone sulfate into testosterone. The authors suggested that RJ-induced improvement of erythropoiesis in their sample could be ascribed to the anabolic effect of testosterone [74].

Menopause is a common inevitable age-related phenomenon that results from reduction of estrogen production in women at an age range of 45 to 55 years [91]. It is often associated with various uncomfortable physical and psychological symptoms. Therefore, menopausal women may receive hormone therapy to alleviate discomfort and lower their risk for various serious diseases such as osteoporosis and cardiovascular disorders. However, hormone therapy is associated with increased risk of cancer [92,93]. Several natural alternatives (including RJ) have been under investigation as safer alternatives. Animal studies document that consumption of RJ by ovariectomized rats improved bone strength [94] and prevented bone loss in a fashion similar to the effect of 17β-estradiol [95]. Cell culture models revealed that the role of RJ in bone formation is due to upregulation of procollagen I α1 gene expression [96,97], enhancement of intestinal calcium absorption, and increased bone calcium content [95]. Human studies lend further support to this evidence; pRJ at a dose of 800 mg/day significantly decreased lower back pain in healthy menopausal women [93]. Likewise, daily consumption of 150 mg of RJ for three months exerted cardioprotective effects by improving lipid profile of postmenopausal women [92].

The healthspan-inducing effects of RJ in humans are not only limited to enhancement of physical health but also include improvement of general mental health [74], reduction of anxiety symptoms [93], improvement of mood, and mild cognitive impairment in the elderly (>60 years old) [98,99]. The psychological and neurological effects of RJ reflect improvement of biomarkers of physical health such as cholesterol [98]. Hypercholesterolemia, in particular, is documented to foster aggregation of β-amyloid around neurons, which causes neuronal loss—a characteristic feature of Alzheimer’s disease. Meanwhile, the reduction of plasma lipids induced by RJ has been associated with enhancement of antioxidative capacities, reduction of β-amyloid deposition, and prevention of neuronal damage [100].

## 5. Healthspan and Longevity Enhancing Mechanisms

It is now evident that RJ contains compounds (MRJPs, royalactin, amino acids, 10-HDA, pantothenic acid, Ach) that can modulate major mechanisms of aging. However, the exact mechanism through which RJ may extend lifespan is not well-understood. The mechanisms through which RJ extend lifespan can differ even within single species (Table 1). Several facets through which RJ can function have been proposed (Figure 3).

### 5.1. Insulin-Signaling/insulin like Growth Factor-1 Signaling

The role of lowered insulin signaling/insulin-like growth factor-1 pathway (*IIS*) in longevity is well documented [101,102]. Findings from studies of *C. elegans* show that *IIS* does not only contribute to prolonged of lifespan, but it also promotes healthspan by resisting different types of stress [102]. The underlying mechanisms include improvement of insulin sensitivity and decrease of insulin levels, suppression of inflammation, reduction of adipose tissue, and increase of adiponectin levels [103]. The lifespan-expanding effect of RJ is associated with an interplay of several genes—expressed in some studies mainly by extending the lifespan of the insulin-like receptor *daf-2* mutants. *daf-2* demonstrates a life-expanding effect by regulating the downstream process of dietary restriction signaling (which lowers food-intake) as well as *TOR* signaling by extending the lifespan of the control *unc-24/+* mutants [66]. Given that *daf-2* is a key upstream component of *IIS*, downregulation of *daf-2* stimulates a signaling cascade that leads to phosphorylation and fine tuning of the main downstream transcription factors of *IIS* known to promote lifespan: *DAF-16*—the *C. elegans* counterpart to the mammalian Forkhead Box O transcription factor (*FOXO*), heat shock transcription factor 1 (*HSF-1*), and *SKN-1/NRF2* [66,67,104]. Among all these pathways, RJ exerted its effect by targeting the activity of *DAF-16/FOXO*, which is a key longevity factor in various species ranging from worms to humans [52,67]. The interactions between *DAF-16* and its associated transcriptional coregulator proteins—host cell factor (*HCF-1*), sirtuin homologue (*SIR-2.1*), and *FTT-2* (a 14-3-3 protein)—lead to formation of a protein complex inside cells [67]. Moreover, activation of *DAF-16* promotes its translocation from the cytoplasm to the nucleus, where it stimulates the expression of multiple genes, which regulate processes that promote longevity: DNA repair, autophagy, antioxidant activity, anti-inflammatory activity, stress resistance, and cell proliferation [10,104]. Aging is associated with increased aggregation and insolubility of various RNA granule proteins (e.g., stress granule proteins). Reduction of *daf-2* receptor signaling heightens *DAF-16* activity, which can efficiently extend lifespan in *C. elegans* by preventing the buildup of misfolded proteins that seed the aggregation of insoluble stress granule proteins [105].

### 5.2. The Mechanistic Target of Rapamycin Signaling

The mechanistic target of rapamycin (*mTOR*) pathway is another signaling cascade that contributes to RJ enhancement of lifespan extension [17,39,66]. The underlying mechanism involves suppression of *mTOR* gene expression by MRJPs and freeze-dried RJ [17,39]. The nutrient sensor *mTOR* is a serine/threonine protein kinase that comes in two structurally and functionally distinct forms: *TOR* Complex 1 (*TORC1*) and *TORC2*. Mammalian *TORC1* (*mTORC1*)—not *TORC2*—plays a major role in aging. *mTORC1* comprises three components, the catalytic subunit mammalian *TOR* (*mTOR*), regulatory-associated protein of target of rapamycin (*RAPTOR*), and mammalian lethal SEC13 protein 8 (*mLST8*) [106,107]. Evidence denotes that inhibition of *mTORC1* with rapamycin extends lifespan up to the double in model organisms [4]. *mTORC1* signaling is regulated by endocrine signaling, especially growth factors and *IGF-1*, as well as nutrients e.g., amino acids, lipids, and glucose, in addition to cellular energy and oxygen levels [107,108]. In this respect, 10-HDA supplementation to *C. elegans* extended the lifespan of the control *unc-24/+* mutants but did not extend the lifespan of the long-lived heterozygous mutants in *daf-15*, which encode *RAPTOR* [66]. The anti-aging activity of the protein kinase *mTOR* originates from its ability to function both in a cell autonomous manner and a non-cell autonomous manner to regulate growth, protein translation, ribosomal biogenesis, autophagy, and cellular metabolism in response to both environmental and hormonal signals [106,108]. It is thought that *mTORC1* inhibition restores cellular physiological integrity, and hence delays age-related pathologies. In details, *mTORC1* promotes translation initiation of mRNAs of metabolism-related genes and ribosomal-related proteins via phosphorylation of two main ribosomal proteins: *S6K1* and *S6K2*. Thus, *mTORC1* inhibition is associated with enhanced endogenous protein degradation as well as less aggregation of proteotoxic and oxidative stress wastes, which in turn preserve homeostasis in the face of oxidative damage. Autophagy genes play a major role in these processes [4,109]. For a detailed description of mechanisms through which *mTOR* functions, we refer readers to these reviews [4,107,108].

### 5.3. Dietary Restriction Signaling

Dietary restriction is another mechanism that promotes longevity in various species. It involves prolonged reduction of intake of most dietary elements except vitamins and minerals (without getting into malnutrition)—it is equivalent to voluntary intermittent fasting in humans, which is reported to prevent numerous debilitating disorders such as abdominal obesity, diabetes, hypertension, and cardiovascular diseases [10]. Recent reviews point out that dietary restriction in humans exhibits similar effects to model organisms in terms of body composition, circulating lipoprotein, inflammatory and metabolite profiles, energy expenditure, and oxidative stress [110,111]. *C. elegans eat-2* mutant is considered a model of dietary restriction; its acetylcholine receptor mutation hinders pharyngeal pumping and limits intake of nutrients [109]. The main lipid of RJ, 10-HDA, which exhibited a lifespan-extending effect did not extend the lifespan of the *eat-2* mutants in *C. elegans*, which denotes that feeding impairment-related dietary restriction signaling was involved in the lifespan extending mechanism. However, progeny production was not delayed in 10-HDA-treated worms—unlike dietary-restricted worms—which indicated that the lifespan-extending effect of 10-HDA was related to the downstream process of the dietary restriction signaling [66]. Indeed, the longevity effect of *eat-2* mutant is mediated by compensatory changes of several energy sensing effectors: dietary restriction downregulates *IIS*, which activates *FOXO*, whereas the energy sensor for cellular AMP/ATP ratio known as *AMPK* gets activated to stimulate catabolic reactions for energy gain by phosphorylating *DAF-16/FOXO*. It also downregulates *AKT/mTOR* through *PHA-4/FOXA* transcription factor and *S6K* (in a *DAF-16/FOXO*-independent manner), and stimulates the expression of autophagy genes—*unc-51/ULK1*, *bec-1/Beclin1*, *vps-34*, *atg-18*, and *atg-7*—which inhibit general protein translation and simultaneously stimulate the translation of specific mRNAs involved in cellular homeostasis [5,10,109]. In addition, dietary restriction causes activation of *NRF2* transcription factor, which suppresses inflammation and counteracts oxidative damage [112].

### 5.4. Epidermal Growth Factor Signaling

RJ, royalactin in particular, extended lifespan of various species by activating the epidermal growth factor receptor (*EGFR*) signaling. As shown in Table 1, RJ activation of *EGFR* signaling involves upregulation of *S6K* and *MAPK*, which results in enhanced locomotor activity and antioxidant capacity; increased *20E* titre, which stimulates growth (increased body size); and increased juvenile titre, which increased fertility—all are effects that indicate healthspan. Royalactin might interact with *LIN-3* to promote the binding of the ligand to the extracellular domain of the *EGFR*, which stimulate *EGF* signaling [68]. Still, the exact mechanism through which RJ affects *EGFR* to promote longevity is not clear. However, it has been recently reported that, royalactin-related *EGFR* signaling induces longevity in *C. elegans* via upregulation of elongation factors and chaperonins, which increase protein translation and proteasome activity—a mechanism that entails rebuilding cellular components and enhancement of cellular detoxification, ribosomal function, and muscle maintenance—rather than stabilizing the existing proteome [113]. Furthermore, activation of *MAPK*—which is stimulated by *EGF*—increases the lifespan of *daf-2* mutants and thus controls the expression of pathogen response genes (*C-type lectins*, *ShK toxins*, and *CUB-like* genes); such increased resistance to pathogens increases lifespan in *C. elegans* [114]. *EGF* pathway functions in a manner that is independent of the *insulin/IGF-like* pathway i.e., activation of its receptor (*LET-23*) is necessary for its action [68]. In addition, it is not affected by deficiency of the *DAF-16/FOXO* transcription factor and it exerts more effects when *daf-2/InR* activity is inhibited [115]. Moreover, royalactin is thought to affect regulators of *EGF* signaling—such as high performance in advanced age genes (*HPA-1* and *HPA-2*)—to stimulate the release of *LIN-3* [68]. In fact, *HPA-1* and *HPA-2* genes negatively regulate *EGF* signaling by binding and sequestering *EGF*. While *HPA-1* is thought to contribute to longevity, *HPA-2* is reported to induce healthspan benefits in *C. elegans* by encoding secreted proteins similar in sequence to extracellular domains of insulin receptor. *EGF* signaling functions via downstream phospholipase C-γplc-3 and inositol-3-phosphate receptor itr-1 to promote healthy aging associated with low lipofuscin levels (age pigments that accumulate during senescence), enhance physical performance, and extend lifespan [115,116]. *EGFR*-mediated antiaging effects seem to be evolutionally conserved from worms to humans. In humans, a recent study tested single-nucleotide polymorphisms (*SNPs*) in *EGFR* for association with longevity. Comparison of genotype frequencies of 41 *EGFR SNPs* between 440 American males of Japanese ancestry aged ≥95 years and 374 men of average lifespan (whites and Koreans) revealed a significant association with longevity for seven *SNPs* in *EGFR*—evidence that genetic variation in *EGFR* contributes to lifespan extension in Japanese people [101].

### 5.5. Oxidative Stress

According to the free radical theory of aging, aging is the result of accumulation of molecular damages that are induced by reactions of free radicals and reactive species that inevitably form during the course of metabolism, which cause errors in cellular processes that are conducive to various age-related disorders [117]. This notion is supported by findings from bee queen studies. Despite the fact that bee queens live longer compared with worker bees, queens enjoy youthful and vigorous cellular functioning. This is attributed to the peroxidation-resistant cell membrane and the lower expression of oxidative stress genes [118,119]. The latter has been attributed, at least in part, to the effect of queen food consumption on microbiota composition and microbiota-derived metabolites, whereas gut microbiota of the short-lived workers—that feast on honey or pollens—is aging i.e., deficient in bacteria that produce metabolites that prevent the expression of oxidative stress genes such as lactobacillus and bifidobacterium [119]. Hence, another possible contributor to the longevity effect of RJ is its enhancement of antioxidation capacity and resistance to oxidative stress, which foster scavenging of free radicals as well as phosphorylation and retention of *DAF-16/FOXO* in the cytoplasm, through the involvement of 3-phosphoinositide-dependent kinase-1. As noted above, RJ supplementation to *C. elegans* increased their resistance to various types of stress [66,67]. Similarly, RJ supplementation to *Drosophila* at low concentrations regulated the gene expression related to oxidative stress and catabolism [16]. In particular, MRJPs fostered the gene expression of *CuZn-SOD*, which was reflected by increased activity of the antioxidant SOD and decreased levels of MDA [12]. Similarly, RJ supplementation to mice decreased kidney DNA and serum levels of 8-hydroxy-2-deoxyguanosine, an age-related marker of oxidative stress [2].

## 6. Discussion

Aging is characterized by progressive functional decline and increased vulnerability to various pathologies, which result from long-term alterations of numerous physiological processes [4]. Therefore, apart from prolonging life, it is necessary to find intelligent anti-aging agents that target age-related genetic pathways and biochemical processes in order to prevent or delay the detrimental effect of age-related physiological changes. Despite the slight discrepancy across studies examined in this review—possibly out of differences in model organisms and experimental designs—the findings indicate that consumption of an appropriate dose of RJ and its ingredients exerts proliferative effects that promote health, increase stress resistance, and prolong lifespan in numerous diverse species. RJ seems to have a potent effect on the functioning of various healthspan and longevity pathways: *IIS*, *mTOR*, *EGFR*, oxidative stress, and dietary restriction eat-2. It might be necessary to further examine the effect of RJ on mechanisms related to microbiota since microbiota composition can interact with pathways of oxidative stress to exert longevity effects [119]. It is also necessary to examine the possibility that such mechanisms may be applicable to humans.

Though several animal models demonstrate extended lifespan as a result of RJ treatment, it is not clear which constituents of RJ are responsible for the longevity effect. MRJPs and royalactin appear to be probable candidates [41,68]. Yet, longevity effects were obtained in many studies that did not include MRJPs, which signifies that other ingredients also enhance health and prolong lifespan. It seems that pRJ [51,67], as well as certain RJ constituents such as RJP_60_ [15]; lipids e.g., 10-HDA [66]; and vitamins e.g., pantothenic acid [54] exhibit healthspan and lifespan-extending activities greater than crude RJ.

For RJ to exert an anti-aging effect, it should be regularly used for long periods of time. Therefore, safety of prolonged use of RJ represents another issue of concern. Bee products such as RJ, honey, pollen, and propolis have long been used as multifunctional substances with various biological activities. Despite the limited possibility of occurrence of allergic reactions, most bee products are relatively nontoxic e.g., high doses of propolis (1400 mg/kg/day) in mice had no effect level [120], whereas oral consumption of RJ in high doses (20 g) in humans produced no adverse effects [86]. Nonetheless, most of the observed positive anti-aging effects of RJ in model organisms were dose-dependent—extremely high doses were associated with unfavorable outcomes in some studies. Yet, with the exception of one study that supplemented bee larvae with RJ 100% [60], the definition of “a high dose” varied between studies and outcomes also varied: no effect [52], increased fertility [60] and survival [64], increased developmental time, and decreased survival [16]. Apart from variation of the studied organisms, as well as nutrient contents of culture media used for in vitro breeding [16,25,41] (which may, in part, explain the discrepancy noticed between these studies), very few attempts were made to explore mechanisms underlying the occurrence of adverse effects when high doses of RJ are used. In this respect, unphysiological exposure of *Drosophila* to high levels of RJ (up to 70%) prolonged development time and shortened lifespan because excess nutrients in high concentrations of RJ altered the expression of genes responsible for the metabolism of amino acids and encoding of glutathione S transferases, which detoxify xenobiotic compounds resulting in distortion of cellular processes [16]. Furthermore, evidence documents that uncontrollably high intake of antioxidants (e.g., vitamin C, vitamin E, N-acetyl cysteine) disturbs the redox balance between processes of oxidation and reduction and induces reductive stress—a shift of body redox levels into an extra reduced state, which may cause severe alterations of cellular functions and lead to pathologies in the same way as oxidative stress [121]. Therefore, adverse effects associated with high doses of RJ may be the result of reductive stress induced by antioxidants in RJ (e.g., phenols, amino acids, peptides, fatty acids, and vitamins). Accordingly, from a cost-effectiveness-oriented point of view, determining an optimal RJ dose would be an important issue in future studies. On the other hand, accumulation of neonicotinoid insecticides and their metabolites in the body and products of honey bees as a result of environmental pollution is associated with occurrence of adverse effects in bees such as oral toxicity, especially during winter—when bee consumption of honey and pollen increases [122]. Nonetheless, several reports indicate that compared with pollen and honey [123,124], RJ contamination with neonicotinoids is very limited (1 to 9.5 µg/kg) [125,126], equivalent to 0.016% of the original concentrations of pesticide fed to bee workers during in vitro breeding [125]. Therefore, RJ maybe considered a safe an anti-aging agent compared with other bee products such as bee pollen.

Preserving the biological activity of RJ is a prerequisite for its use to efficiently promote health. RJ seems to be sensitive to temperature and other methods of handling, which may affect its ingredients and potency e.g., RJ lost most of its bioactive components after 30 days of storage at 40 °C [41] and pRJ had a better effect than crude RJ [52]. Further, the effect of commercial RJ was suboptimal compared with RJ from reliable sources [16]. Another concern is the effect of digestive enzymes on bioactive ingredients of RJ when it is orally ingested. For instance, MRJPs prevent senescence of human cells in vitro [39]; however, MRJPs are rapidly digested in the stomach and small intestine except for MRJP2, which can remain in the intestine as a full-length protein for 40 min and it should be resorbed quickly were it to produce any biological effect [127].

Genetic variation is another challenge if we are to identify candidates for pathology prevention in humans. RJ as a dietary supplement may prevent some of the main age-related diseases. However, various types of genetic variation may affect response to RJ treatment. Taking gender as an example, RJ increased the body size of females only in some species such as silkworms [65]—an effect of increased levels of fertility hormones [17,41]. In humans, women live longer than men, yet they have higher genetic risk for some age-related diseases (e.g., Alzheimer’s disease) than men, which indicates the possibility for gender-specific treatment targets [128]. Other types of genetic variation are also important; for instance, RJ affects the *EGFR* pathway, which has been shown to be associated with prolonged lifespan in Japanese—but not in whites or Koreans—which highlights a role of ethnic difference in genotype and epigenotype [101]. Other factors, such as other genetic variation factors, diet, and activity level (which affects healthspan and longevity) should be considered in future clinical trials.

## 7. Conclusions

Accumulating evidence from studies of honey bees, fruit flies, crickets, silkworms, mice, and humans indicates that RJ has an obvious role in modulating the mechanisms of aging, which can promote healthspan and longevity. Although most studies relate the anti-aging properties of RJ to MRJPs and 10-HDA, it is not exactly clear which fractions or doses are most beneficial. The discovery of predictive biomarkers that take into account individual variation in genotype and epigenotype is necessary in order to conduct sound clinical trials that can test the efficacy of RJ on the rate of biological aging as well as the risk of age-related diseases.

## Figures and Tables

**Figure 1 ijms-20-04662-f001:**
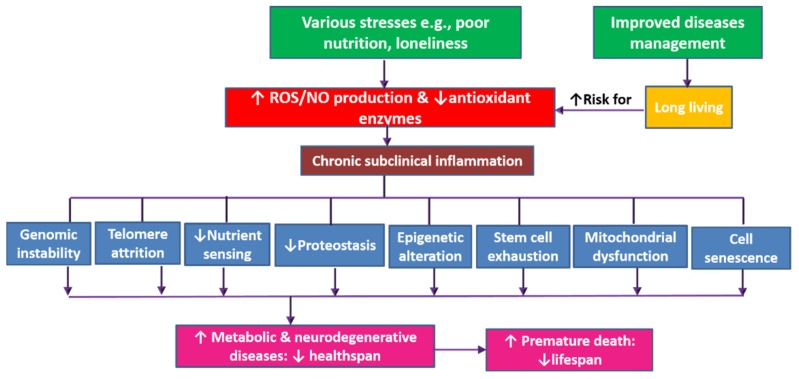
The dynamics of aging-related decrease of healthspan and lifespan. The production of free radicals increases with aging (which is on the rise out of improved disease management) and also with exposure to various stresses (e.g., pollution, poor nutrition, and psychological stress) leading to oxidative stress. This is associated with reduced production of antioxidant enzymes and activation of inflammatory pathways. Both oxidative stress and chronic inflammation lead to a trail of cellular and molecular alterations: telomere attrition, epigenetic alterations, genome instability, reduced proteostasis, disturbed nutrient sensing, mitochondrial dysfunction, cellular senescence, stem cell exhaustion, and altered intercellular communication. Such age-related physiological alterations result in poor health and loss of function out of increased occurrence of various metabolic and neurodegenerative disorders. Age-related disorders are associated with increased mortality and premature death i.e., decreased lifespan.

**Figure 2 ijms-20-04662-f002:**
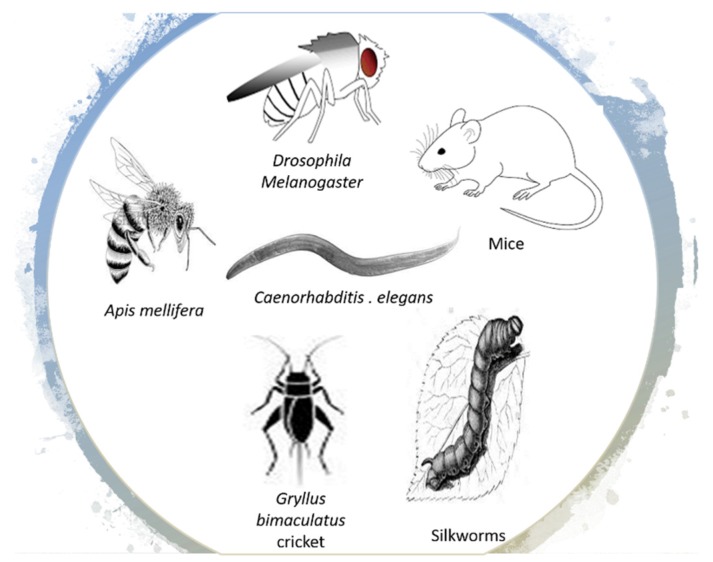
Royal jelly and its components improve healthspan and extend lifespan in different species.

**Figure 3 ijms-20-04662-f003:**
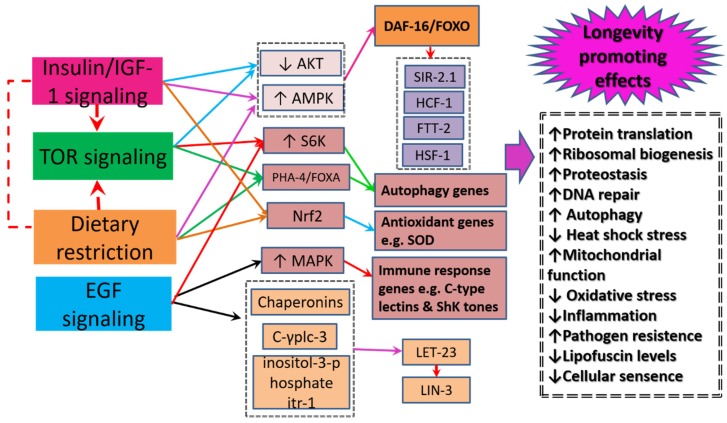
Probable mechanisms through which royal jelly (RJ) and its components extend lifespan. RJ reduces insulin/insulin-like growth factor-1 signaling (*IIS*), which heightens the activity of *DAF-16*—the *C. elegans* counterpart to the mammalian Forkhead Box O transcription factor (*FOXO*). Activation of *DAF-16/FOXO* promotes its translocation from the cytoplasm to the nucleus and fosters interactions with its associated transcriptional co-regulator proteins: host cell factor (*HCF-1*), sirtuin homologue (*SIR-2.1*), and *FTT-2* (a 14-3-3 protein). This results in formation of protein complex inside cells and enhancement of the expression of multiple longevity promoting genes as well as regulation of downstream process of dietary restriction signaling (which lowers food-intake) and the mechanistic target of rapamycin (*mTOR*) pathway by extending the lifespan of the control *unc-24/+* mutants. The interplay among several genes involved in *IIS*, *mTOR*, and dietary restriction signaling boosts key longevity-related cellular processes: DNA repair, autophagy, antioxidant activity, anti-inflammatory activity, stress resistance, and cell proliferation. On the other hand, RJ activates epidermal growth factor receptor (*EGFR*) signaling mainly by activating its receptor (*LET-23*). *EGFR* functions via downstream phospholipase C-γplc-3 and inositol-3-phosphate receptor itr-1 to upregulate elongation factors and chaperonins, which increase protein translation and proteasome activity—a mechanism that entails rebuilding cellular components; enhancement of cellular detoxification, ribosomal function, and muscle maintenance; and reduction of lipofuscin levels (age-pigments that accumulate during senescence).

**Table 1 ijms-20-04662-t001:** Effect of Royal jelly and its components on lifespan and healthspan in different species.

RJ/RJ Components	Species	Effect on Lifespan	Effect on Healthspan	Main Mechanism of Action	References
RJ	*Apis mellifera* (worker)	No effect on lifespan	↑ Ovarian activation	NI—Inability to defecate in 100% RJ-fed bees	[60]
RJ	*Apis mellifera* (worker)	↑ Mean lifespan	↑ Ovarian activation	NI	[63]
RJ	*Apis mellifera* (worker)	--	↑ Ovarian activation	↑ Brain levels of tyrosine, dopamine, and tyramine	[61]
RJ + Ach	*Apis mellifera* (larvae)	↑ Mean lifespan	--	NI—possibly the trophic effects of Ach mediated via muscarinic or nicotinic receptors	[56]
RJ RJP_60_	*Apis mellifera* (worker)	↑ Mean and maximum lifespan	--	NI Lower DNA methylation levels	[15]
RJ	*Apis mellifera* (worker)	↑ Mean lifespan	↑ Expression of memory genes	NI	[62]
RJ, heat-treated RJ, pKRJ, RJ plus MRJP1 vs RJ plus MRJPs2,3,5	*Apis mellifera* (larvae)	↑ Mean lifespan (except MRJP1)	↑ Ovarian activation	NI	[25]
Dehydrated RJ RJ pantothenic acid RJ organic acids	*Drosophila M.*	↑ Mean lifespan	--	NI Synergizing action of other vitamins	[54]
RJ	*Drosophila M.*	↑ Mean lifespan (both sexes)	--	↑ Anti-oxidation capacity--SOD and CAT levels	[64]
RJ, royalactin	*Drosophila Canton-S* *Apis mellifera* (larvae)	↑ Mean lifespan	↑ Body size ↑ Cell size ↑ Eggs laying ↓ Developmental time	↑ *EGFR*-mediated signaling pathway, *S6K*, *MAPK*, juvenile hormone titre, *20E* titre	[41]
Freeze-dried RJ	*Drosophila M.*	↑ Mean lifespan (males only)	↓ Developmental time ↑ Eggs laying	↓ *Insulin/IGF-1* (*dilp5*) signaling ↓ Target of Rapamycin signaling	[17]
RJ	*Drosophila Canton S*	No effect on lifespan	↑ Body size	↑ Gene expression related to oxidative stress and catabolism	[16]
MRJPs	*Drosophila M.*	↑ Mean and maximum lifespan (both sexes)	↑ Feeding ↑ Eggs laying	↑ Anti-oxidation capacity—*CuZn-SOD* signaling ↑ *EGFR*-mediated signaling	[12]
RJ	*Gryllus bimaculatus* crickets silkworms	↑ Mean lifespan in crickets (both sexes)	↑ Eggs size (silkworms) ↑ Body size ↓ Developmental time,	NI	[65]
RJ, pRJ, pRJ-Fr.5, 10-HDA	*C. elegans*	↑ Mean lifespan	--	↓ *Insulin/IGF-1* and *ins-9* signaling	[52]
Royalactin	*C. elegans*	↑ Mean lifespan	↑ Locomotion in early and mid-adulthood	↑ *EGFR*-mediated signaling pathway	[68]
10-HDA	*C. elegans*	↑ Mean lifespan	↑ Stress resistance	↓ *Insulin/IGF-1* signaling ↓ Target of Rapamycin signaling ↑ Dietary Restriction	[66]
RJ, pRJ	*C. elegans*	↑ Mean lifespan	↑ Stress resistance	↓ *Insulin/IGF-1* signaling	[67]
Powdered RJ	C3H/HeJ mice	↑ Mean lifespan	--	↓ Oxidative stress and DNA damage	[2]
RJ	Male Swiss albino mice	↑ Survival time after NaNO2 IP injection	--	NI	[71]
RJ, pRJ	D-galactose induced aging mice model	--	↓ Atrophy of thymus ↓ Weight loss ↓ Locomotor decline ↑ Learning and memory	↓ Oxidative stress	[69]
RJ, pRJ	HET mice	No effect on lifespan	↓ Muscle atrophy ↓ Age-related motor impairment	↑ Muscle satellite cell (muscle stem cell) markers Suppression of catabolic genes	[70]

**Abbreviations**. RJ: royal jelly; ↑: increase; ↓: decrease; NI: not investigated; Ach: acetylcholine; RJP_60_: RJ protein attained by precipitation with 60% ammonium sulfate; EGRF: epidermal growth factor receptor; MRJPs: major royal jelly proteins; pKRJ: protease K-treated RJ; SOD: superoxide dismutase; CAT: catalase; *20E*: 20-hydroxyecdysone; pRJ: protease-treated RJ; pRJ-Fr.5: protease-treated-fraction 5; 10-HDA: 10-hydroxy-2-decenoic acid; IP: intraperitoneal; HET: genetically heterogeneous head tilt.

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
