# Peer review of "Royal Jelly and Its Components Promote Healthy Aging and Longevity: From Animal Models to Humans"

_ijms, 2019, doi:10.3390/ijms20194662_

Round 1

Reviewer 1 Report

Review of the article: „ Royal jelly and its components promote healthy aging and longevity: From animal models to humans” – revised version

Manuscript ID - ijms-597754

I have compared the revised version of the manuscript with the previous one, and I have to admit that the new document is much better. The authors considered all my comments and suggestions. First of all the authors presented detailed description of chemical composition of RJ. In my opinion the current version of the manuscript can be accepted. I have only several, minor of importance, comments.

Detailed comments:

The new version of title is better.

Abstract is well presented, the authors have taken into account all my comments

The paragraph 2 - Chemical composition of RJ is very well prepared. However I have some suggestions which the authors can consider preparing final version of the publication.

Lines 91-95 the sentence: “RJ sugar content is extremely high compared with WJ …” is very long and not really clear – please try to rephrase it, the abbreviation WJ must be explained

Line 108 - … human cancer colon cells please check if bacteria are eliminated only from cancer cells

Lines 172-173 GAE and RE should be explained (these are equivalents – it should be clearly written).

All these comments are of minor importance. In my opinion the next revision (by reviewer) of the manuscript is not necessary.

Final opinion – minor revision

Author Response

Response to Comments of Reviewer 1

We appreciate Reviewer 1’s help and concerns for clarity as indicated by the provided comments. The comments are addressed line-by-line as shown below. Replies come underneath in red according to the example available on your website, and modifications in the text are highlighted in red to make it easier to grasp.

Lines 91-95 the sentence: “RJ sugar content is extremely high compared with WJ …” is very long and not really clear – please try to rephrase it, the abbreviation WJ must be explained

Response 1: The sentence was rephrased into three simpler sentences to clarify its meaning as the reviewer suggested.

WJ stands for worker jelly, it has been defined previously (highlighted text, line 69). Therefore, we used the abbreviation in the following instances reminding the reader again of what WJ refers to (lines 92 and 93).

Line 108 - … human cancer colon cells please check if bacteria are eliminated only from cancer cells

Response 2: The reported study used human colon cancer cells (WiDr) for all experiments. Though, we could not locate other reports of the antimicrobial activity of 10-HDA in humans, we reported its effect in vitro and in vivo in mice (lines 109 and 110).

Lines 172-173 GAE and RE should be explained (these are equivalents – it should be clearly written).

Response 3: GAE and RE have been clearly defined (line 177). 

We hope that the comments were properly handled and that the revised version will be suitable for the publication.

Best regards,

Reviewer 2 Report

In their Review the authors had stated that higher concentrations (40-70%) of RJ had adverse effects in Drospohila (rows 305-307 of the manuscript) but they did not provide any possible explanation for that.

So my questions are:

Can authors please provide possible explanations of the fact that higher concentrations of RJ can results in adverse effects? What are the posibble mechanisms behind this adverse effects? Please add these explanations into your review.

Can authors please comment about influence of other bee products in higher concentrations such as propolis or bee pollen on health and well-being and add this into the review? Can authors also please comment are the mechanisms behind the adverse affects of high concentrations of RJ similar or maybe the same in situations with the addition of other bee products such as propolis or bee pollen and add this also into their review?

Overall, my opinion is that revised version of this review offers many important information about RJ, its application and potential health effects, with specially detailed emphasis of RJ influence on aging and in my opinion this review should be published after the authors add small explanations that I had ask for before.

Sincerely,

Author Response

Response to Comments of Reviewer 2

First of all, we need to express our gratitude to Reviewer 2 for these helpful comments. The comments are addressed line-by-line as shown below. Replies come underneath in red according to the example available on your website, and modifications in the text are highlighted in red to make it easier to grasp.

In their Review the authors had stated that higher concentrations (40-70%) of RJ had adverse effects in Drospohila (rows 305-307 of the manuscript) but they did not provide any possible explanation for that.

Response 1: An explanation for the adverse effect associated with high concentrations of RJ is provided in lines 312-214.

Can authors please provide possible explanations of the fact that higher concentrations of RJ can results in adverse effects? What are the posibble mechanisms behind this adverse effects? Please add these explanations into your review.

Response 2: We totally agree with Reviewer 2, determining a safe and effective dose is extremely important. Although the term “higher concentrations” has been exclusively used in most of the reviewed studies, and also in the current review, there is no gold standard to tell how high is very high. In addition, various model organisms were studied and the content of culture media, apart from RJ treatment, also varied which might confound the results. More, among all the reviewed studies, only one study tended to sensibly investigate the mechanism through which RJ produced adverse effects in Drosophila. Based on the reviewer’s comment, we have added a paragraph in the Discussion (lines 627-660). In this paragraph, we briefly reported on outcomes of high RJ dosage, which were not all unfavourable, and attempted to explore the mechanisms underlying adverse effects in high dosage such as altered metabolism of amino acids and possibly reductive stress.

Can authors please comment about influence of other bee products in higher concentrations such as propolis or bee pollen on health and well-being and add this into the review? Can authors also please comment are the mechanisms behind the adverse affects of high concentrations of RJ similar or maybe the same in situations with the addition of other bee products such as propolis or bee pollen and add this also into their review?

Response 3: In the same paragraph added in the discussion we have reported on high dosage of propolis in mice as a mammal model and on high dose of RJ in humans and results are consistent—both were non-toxic (lines 628-632). Even more, RJ content of pesticides as a result of environmental pollution is even lower than in pollens and honey i.e., it may be safer than other bee products (lines 653-660).

In final, we really appreciate all your valuable time and efforts. We hope that comments were satisfactorily addressed and that the revised version will be suitable for the publication.

Best regards,
